# PRIVATELY COUNTING PARTIALLY ORDERED DATA

**Matthew Joseph, Mónica Ribero & Alexander Yu**
Google Research NY *

## ABSTRACT

We consider differentially private counting when each data point consists of $d$ bits satisfying a partial order. Our main technical contribution is a problem-specific $K$-norm mechanism that runs in time $O(d^2)$. Experiments show that, depending on the partial order in question, our solution dominates existing pure differentially private mechanisms, and can reduce their error by an order of magnitude or more.

## 1 INTRODUCTION

A differentially private (DP) (Dwork et al., 2006) statistic incorporates randomness to obscure the contribution of any single data point. To achieve this, differentially private algorithms typically require — and, if necessary, enforce — restrictions on the data points. A common restriction is $\ell_p$ sensitivity: if the statistic $T$ is a vector in $\mathbb{R}^d$, adding a user's data point must not change the $\ell_p$ norm of $T$ by more than $\Delta_p$. This framework is generic enough to apply to a wide variety of problems, but it can yield relatively poor performance for statistics whose sensitivity is not tightly characterized by an $\ell_p$ norm. One such statistic is counting on *partially ordered data*.

As a straightforward running example, the National Health Interview Survey (Services & Medicaid, 2024) has been administered annually since 1957 by the Center for Disease Control. The Hypertension section of the 2024 survey first asks if the respondent has been told they have hypertension and then asks if they have been told multiple times. The survey structure requires a respondent to answer "yes" to the first question in order to answer "yes" to the second; equivalently, if the answers are given as binary vector $x$, the survey structure imposes partial order $x_2 \leq x_1$. A count of this partially ordered data then records the number of "yes" responses to each question. Other examples of partially ordered data include software library dependencies, coursework prerequisites, and any data encoded using directed acyclic graphs.

It is possible to apply $\ell_p$ sensitivity to partially ordered data. In the survey example, a respondent can answer "yes" to every question, so the vector of counts has $\ell_1$ sensitivity $d$. However, this analysis also protects against inputs that cannot occur, as the binary and order constraints means that no response can have the form $(d, 0, \ldots, 0)$ or $(0, 1, \ldots)$. In this sense, straightforward application of Laplace (or other $\ell_p$ mechanism) noise may be "loose".

### 1.1 CONTRIBUTIONS

Our main technical contribution is an efficient implementation of the $K$-norm mechanism (Hardt & Talwar, 2010) for counting partially ordered data. Our solution can be applied to any partial order, satisfies pure differential privacy, and runs in time $O(d^2)$ (Theorem 3.15). This provides a more than cubic speedup over the fastest general polytope sampler and is provably exponentially faster than any $\ell_p$ ball rejection sampler (Theorem 3.17). Experiments using both synthetic and real-world partial orders demonstrate significant error reductions over existing DP mechanisms (Section 4).

### 1.2 RELATED WORK

Joseph & Yu (2024) previously constructed efficient implementations of the $K$-norm mechanism to obtain better noise distributions for the problems of (contribution-bounded) sum, count, and vote. For a longer discussion of the $K$-norm mechanism, its variants, and its relation to other query-answering solutions like the projection (Nikolov et al., 2013; Nikolov, 2023), matrix (Li et al.,

---

*{mtjoseph, mribero, alexjyu}@google.com.

2015; McKenna et al., 2018), and factorization (Edmonds et al., 2020; Nikolov & Tang, 2023) mechanisms, we refer the interested reader to their paper; here, we note that these mechanisms can be viewed as general but possibly slow approximations of optimal constructions of problem-specific private additive noise. In contrast, we provide a problem-specific but optimal and fast construction.

The simplest difference between our work and that of Joseph & Yu (2024) is that we consider different problems that require different sampling techniques. In more detail, these different problems are characterized by different combinatorial objects. In Joseph & Yu (2024), their sum and vote polytopes are based on a truncated hypercube and permutohedron, respectively. Here, the poset polytope is most closely related to the double poset polytope. Similarly, though our samplers are also based on identifying an efficiently sampleable triangulation of the polytope, the triangulations themselves are different: the sum polytope triangulation is indexed by permutations with a fixed number of ascents, while ours is indexed by pairs of non-interfering chains in the Birkhoff lattice.

## 2 PRELIMINARIES

### 2.1 DIFFERENTIAL PRIVACY

The material in this subsection is largely adapted from the preliminaries of Joseph & Yu (2024), who derived efficient instances of the $K$-norm mechanism for different problems. We use pure differential privacy, in the add-remove model.

**Definition 2.1** (Dwork et al. (2006)). *Databases $X, X'$ from data domain $\mathcal{X}$ are neighbors $X \sim X'$ if they differ in the presence or absence of a single record. A randomized mechanism $\mathcal{M} : \mathcal{X} \to \mathcal{O}$ is $\varepsilon$-differentially private (DP) if for all $X \sim X' \in \mathcal{X}$ and any $S \subseteq \mathcal{O}$,*

$$\mathbb{P}_{\mathcal{M}} \left[ \mathcal{M}(X) \in S \right] \leq e^{\varepsilon} \mathbb{P}_{\mathcal{M}} \left[ \mathcal{M}(X') \in S \right].$$

Our solution will apply the $K$-norm mechanism.

**Lemma 2.2** (Hardt & Talwar (2010)). *Given statistic $T$ with $\| \cdot \|$-sensitivity $\Delta$ and database $X$, the $K$-norm mechanism has output density $f_X(y) \propto \exp\left(-\frac{\varepsilon}{\Delta} \cdot \|y - T(X)\|\right)$ and satisfies $\varepsilon$-DP.*

We apply an instance of the $K$-norm mechanism chosen using our statistic's *sensitivity space*.

**Definition 2.3** (Kattis & Nikolov (2017); Awan & Slavković (2021)). *The* sensitivity space *of statistic $T$ is $S(T) = \{T(X) - T(X') \mid X, X'$ are neighboring databases$\}$.*

Sensitivity spaces that induce norms are particularly interesting, as the corresponding $K$-norm mechanisms enjoy certain notions of optimality.

**Lemma 2.4.** *If set $W$ is convex, bounded, absorbing (for every $u \in \mathbb{R}^d$, there exists $c > 0$ such that $u \in cW$), and symmetric around 0 ($u \in W \Leftrightarrow -u \in W$), then the function $\| \cdot \|_W : \mathbb{R}^d \to \mathbb{R}_{\geq 0}$ given by $\|u\|_W = \inf\{c \in \mathbb{R}_{\geq 0} \mid u \in cW\}$ is a norm, and we say $W$ induces $\| \cdot \|_W$.*

**Definition 2.5.** *If statistic $T$ has a sensitivity space convex hull $CH(S(T))$ satisfying Lemma 2.4, we say $T$ induces a norm, and call $CH(S(T))$ the induced norm ball of $T$.*

Awan & Slavković (2021) show that the instance of the $K$-norm mechanism using a statistic's induced norm is optimal with respect to entropy and conditional variance (see Sections 3.2 and 3.3 of their paper for details). The only remaining challenge is running the resulting mechanism.

**Lemma 2.6** (Hardt & Talwar (2010)). *Running the $K$-norm mechanism reduces to sampling the unit ball for the norm $\| \cdot \|$.*

Our main technical contribution is therefore a fast sampler for the induced norm ball for counting partially ordered data. This is a $d$-dimensional polytope with $\Omega(d)$ constraints. Applying the best general sampler takes time $\tilde{O}(d^{2+\omega})$ where $\omega \geq 2$ is the matrix multiplication exponent (Theorem 1.5 of Laddha et al. (2020)), and achieving a close enough approximation for $O(\varepsilon)$-DP adds another $O(d)$ factor (Appendix A of Hardt & Talwar (2010)).

### 2.2 PARTIALLY ORDERED DATA

Throughout, we assume that our goal is to compute a sum $T(X) = \sum_{x \in X} x$ of length-$d$ binary vectors with bits that satisfy a partial order.

**Definition 2.7.** *A* partially ordered set *or* poset $(P, \preceq)$ *is a finite set $P$ together with a reflexive, transitive, and anti-symmetric relation $\preceq$. If $p_1, p_2 \in P$ and $p_1 \preceq p_2$, we say that $p_1$ is a* child *of $p_2$ and $p_2$ is a* parent *of $p_1$. A linear extension* of poset $(P, \preceq)$ *is a total ordering $(P, \preceq^*)$ that preserves $\preceq$, i.e. an ordering of the elements of $P$ given by $p_1 \preceq^* p_2 \preceq^* \ldots \preceq^* p_d$ such that $p_i \preceq p_j$ implies $i < j$. We assume that a poset is given in the form of a $|P| \times |P|$ binary matrix $M$ where $M_{ij} = 1 \Leftrightarrow p_i \preceq p_j$. We say a length-$|P|$ binary vector $x$ is* partially ordered *if $p_i \preceq p_j$ implies $x_i \leq x_j$.*

We also assume that each poset has a single root.

**Assumption 2.8.** *The poset $(P, \preceq)$ has a root $r \in P$ such that $p \preceq r$ for all $p \in P$.*

For surveys, this corresponds to a question asking if the respondent wants to take the survey at all, and allows us to count nonrespondents. Our mechanism is optimal for these posets in the sense described by Awan & Slavković (2021). Note also that if we are instead given a poset without a root, we can simply add a root, apply our mechanism, and ignore the added dimension in the final output, as done in our experiments. By post-processing, this does not alter the privacy guarantee.

## 3 AN EFFICIENT MECHANISM

We start with a high level summary of the sampler. Since our goal is to sample the norm ball induced by our statistic, we first show that counting partially ordered data induces a norm, and that the induced norm ball can be expressed as a combinatorial object called a double order polytope (Section 3.1). This connection allows us to leverage a triangulation for double order polytopes developed by Chappell et al. (2017). The problem then reduces to sampling a single simplex from the triangulation. However, the simplices in this triangulation are indexed by pairs of non-interfering chains (Definition 3.10) – essentially disjoint sets of elements in the poset ground set with a special structure given by the order relation – and it is not obvious how to sample such an object uniformly at random. To do so, we prove a bijection between the pairs of non-interfering chains of the triangulation and a family of new structures called extended bipartitions (Definition 3.12), for which we construct a uniform sampler (Section 3.3). Putting these elements together yields an overall sampler of a simplex of the triangulation, from which we can easily sample a single point (Section 3.4).

### 3.1 POSET BALL

As described in Section 2.1, an efficient application of the optimal $K$-norm mechanism for counting partially ordered data reduces to sampling the norm ball induced by its sensitivity space. We start with some basic results about the structure of this norm ball. Throughout, we denote our original poset by $P^*$. Recall from Assumption 2.8 that we assume the poset $(P^*, \preceq)$ ordering our data points has a root $r$ such that $p \preceq r$ for all $p \in P^*$; we omit this assumption for the rest of this section.

**Lemma 3.1.** *Let $T_{(P^*, \preceq)}(X) = \sum_{x \in X} x$ be a statistic summing data points $x$ that satisfy a partial order $(P^*, \preceq)$ (Definition 2.7). Then $T_{(P^*, \preceq)}$ induces a norm.*

*Proof.* We verify the conditions in Lemma 2.4. We write $T = T_{(P^*, \preceq)}$ for brevity.

$CH(S(T))$ is a convex hull and so immediately convex. Let $V(CH(S(T)))$ be the set of vertices of $CH(S(T))$. By the definition of sensitivity space, the vertices of $CH(S(T))$ are the length-$|P^*|$ partially ordered binary vectors and their negations. There are finitely many vertices, so $CH(S(T))$ is bounded. Any point $p \in CH(S(T))$ is a convex combination of the vertices $\sum c_i v_i$, and $v_i \in V(CH(S(T)))$ implies $-v_i \in V(CH(S(T)))$, so $-p = \sum c_i (-v_i) \in CH(S(T))$. Thus, $CH(S(T))$ is symmetric around the origin.

It remains to show $CH(S(T))$ is absorbing. Consider any linear extension $(P^*, \preceq')$ of $(P^*, \preceq)$. Then $(P^*, \preceq')$ has all of the relations in $(P^*, \preceq)$, so $S(T_{(P^*, \preceq')}) \subset S(T_{(P^*, \preceq)})$, and $CH(S(T_{(P^*, \preceq')})) \subset CH(S(T_{(P^*, \preceq)}))$. Define $\{v_1, \ldots, v_d\}$ to be the set of binary vectors where $v_i$ is the vector whose first $i$ entries are 1 while the rest are 0. Then $V(CH(S(T_{(P^*, \preceq')})))$ is $\{\pm v_1, \ldots, \pm v_d\}$. Since $\{v_1, \ldots, v_d\}$ forms a basis of $\mathbb{R}^d$, there is an invertible linear map $A$ mapping $\{v_1, \ldots, v_d\}$ to the standard basis. In particular, $A(CH(S(T_{(P^*, \preceq')})))$ is the unit $\ell_1$ ball $B_1^d$. Let $b$ be a small origin centered ball around 0 in $B_1^d$. Then $A^{-1}(b)$ is a small origin

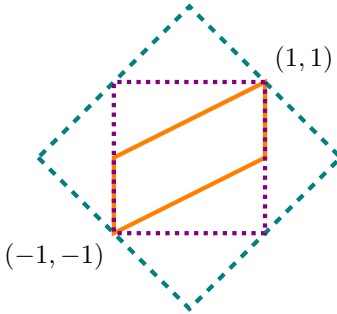

Figure 1: The solid orange border outlines the $d = 2$ path graph ($x_2 \leq x_1$) poset ball. The dashed teal border is the minimum containing $\ell_1$ ball, and the dotted purple border is the minimum containing $\ell_\infty$ ball.

centered ellipsoid in $CH(S(T_{(P^*, \preceq')}))$, i.e. $0$ is an interior point of $CH(S(T_{(P^*, \preceq')}))$. Since $S(T_{(P^*, \preceq')}) \subset S(T_{(P^*, \preceq)})$, it follows that $CH(S(T_{(P^*, \preceq)}))$ is absorbing. $\qquad\square$

We shorthand $T$'s induced norm ball as the *poset ball*. As a simple example, the poset ball for the path graph $x_2 \leq x_1$ appears in Figure 1. The next result shows that the poset ball can be reinterpreted as an object from combinatorics, the double order polytope. This interpretation allows us to apply results about the double order polytope from Chappell et al. (2017).

Reasoning about the double order polytope requires some additional definitions related to posets.

**Definition 3.2.** *A* double poset *is a triple* $(P, \preceq_+, \preceq_-)$ *where* $P_+ = (P, \preceq_+)$ *and* $P_- = (P, \preceq_-)$ *are posets on the same ground set. A double poset is* compatible *if* $P_+$ *and* $P_-$ *have a shared linear extension.*

**Definition 3.3** (Chappell et al. (2017)). *Let* $\mathcal{P}_1, \mathcal{P}_2 \subset \mathbb{R}^d$ *be polytopes. The* Cayley sum *of* $\mathcal{P}_1$ *and* $\mathcal{P}_2$ *is defined by* $\mathcal{P}_1 \boxplus \mathcal{P}_2 = CH(\{1\} \oplus \mathcal{P}_1 \cup \{-1\} \oplus \mathcal{P}_2)$. *Define* $\mathcal{P}_1 \boxminus \mathcal{P}_2 = \mathcal{P}_1 \boxplus -\mathcal{P}_2$. *The* order polytope $\mathcal{O}(P)$ *of a poset* $(P, \preceq)$ *is the set of all order preserving functions* $f : P \to [0, 1]$ *such that* $0 \leq f(a) \leq f(b) \leq 1$ *for all* $a, b \in P$ *where* $a \preceq b$. *The* double order polytope $\mathcal{O}_2(P)$ *of a double poset* $(P, \preceq_+, \preceq_-)$ *is* $\mathcal{O}(P_+) \boxminus \mathcal{O}(P_-)$.[1]

We can now state the connection between the poset ball and double order polytope.

**Lemma 3.4.** *The poset ball for poset* $(P^*, \preceq)$ *is* $\mathcal{O}_2(P^* - r)$, *the double order polytope on the double order poset* $(P^* - r, \preceq, \preceq)$.

*Proof.* Let $S$ be the set of functions $f : P^* \to \{0, 1\}$ such that $0 \leq f(a) \leq f(b) \leq 1$ for all $a, b \in P^*$ where $a \preceq b$. We can view functions in $S$ as $|P^*|$-dim binary vectors in the natural way. Recall that the poset ball is $CH(S \cup -S)$. Equivalently, it is $CH(\mathcal{O}(P^*) \cup -\mathcal{O}(P^*))$. We apply the following general result about the convex hulls of reflected polytopes. A similar result is stated without proof in Chappell et al. (2017); for completeness, a proof appears in Appendix A.

**Claim 3.5.** *Let* $\mathcal{P}$ *be a* $(d - 1)$-*dim convex polytope in* $\mathbb{R}^d$ *lying in the hyperplane* $x_1 = 1$. *Let* $\mathcal{P}' = \mathcal{P} \cup \{0\}$. *Then* $V(CH(\mathcal{P}' \cup -\mathcal{P}')) = V(\mathcal{P}) \cup V(-\mathcal{P})$.

By Claim 3.5, the vertices of $CH(\mathcal{O}(P^*) \cup -\mathcal{O}(P^*))$ are $V(\mathcal{O}(P^*)) \cup V(-\mathcal{O}(P^*)) - \{0\}$. The vertices in $V(\mathcal{O}(P^*)) - \{0\}$ are of the form $1 \oplus v$ where $v$ is a vertex of $V(\mathcal{O}(P^* - r))$, and the first dimension corresponds to the $r$-dimension. In other words, the vertices of this shape are exactly the vertices of $\mathcal{O}_2(P^* - r)$, the double order polytope on the double poset $(P^* - r, \preceq, \preceq)$. $\qquad\square$

By Lemma 3.4, our new goal is to construct a fast sampler for the double order polytope $\mathcal{O}_2(P^* - r)$.

---

[1]This is a slight modification of the definition of the double order polytope given by Chappell et al. (2017), but the change does not meaningfully affect the properties of this structure (see Remark A.1 for details).

### 3.2 DOUBLE ORDER POLYTOPES AND DOUBLE CHAIN POLYTOPES

We sample $\mathcal{O}_2(P^* - r)$ by constructing a triangulation that decomposes it into disjoint $d$-dimensional simplices of equal volume. By this construction, it suffices to sample one of the simplices uniformly at random and then return a uniform random sample from the chosen simplex. The following result about uniformly sampling a simplex is folklore; we take the statement from Joseph & Yu (2024).

**Lemma 3.6.** *A collection of points* $x_0, \ldots, x_d \in \mathbb{R}^n$ *with* $n \geq d$ *are* affinely independent *if* $\sum_{i=0}^{d} \alpha_i x_i = 0$ *and* $\sum_{i=0}^{d} \alpha_i = 0$ *implies* $\alpha = 0$. *A $d$-simplex is the convex hull of $d + 1$ affinely independent points and can be uniformly sampled in time* $O(d \log(d))$.

Since sampling a simplex is easy, it remains to find a triangulation of $\mathcal{O}_2(P^* - r)$ into simplices. For completeness, the following definition formalizes the notion of a triangulation.

**Definition 3.7.** *Let* $\mathcal{P} \subset \mathbb{R}^d$ *be a polytope. For $0 \leq k \leq d - 1$, a proper $k$-dim face of $\mathcal{P}$ is an intersection $I$ of the form $H \cap \partial \mathcal{P}$ of a $(d-1)$-dim hyperplane $H$ with the boundary of $\mathcal{P}$, such that the $I \subset \partial \mathcal{P}$ and $\dim(I) = k$. A* subdivision *of $\mathcal{P}$ is a collection of $d$-dim polytopes $P_1, ..., P_m$ such that $\mathcal{P} = \cup_{i=1}^{m} P_i$ and for each $1 \leq i < j \leq m$ the (possibly empty) set $P_i \cap P_j$ is a proper face of each of $P_i$ and $P_j$. A* triangulation *of $\mathcal{P}$ is a subdivision containing only simplices. A simplex with vertices in a lattice $\Lambda$ is* unimodular *if it has minimal volume. A triangulation is unimodular if all of its simplices are unimodular.*

We use a result from Chappell et al. (2017) that provides a mapping between the double order polytope and a related geometric object, the double chain polytope.

**Definition 3.8** (Chappell et al. (2017)). *Given poset $P$, a* chain *is a sequence of (strictly) increasing elements in $P$. The* chain polytope $\mathcal{C}(P)$ *is the set of functions $g : P \to \mathbb{R}^{\geq 0}$ such that $g(a_1) + ... + g(a_k) \leq 1$ for all chains $a_1 \preceq ... \preceq a_k$ of $P$. The* double chain polytope $\mathcal{C}_2(P)$ *of a double poset $(P, \preceq_+, \preceq_-)$ is $\mathcal{C}(P_+) \boxminus \mathcal{C}(P_-)$*

**Lemma 3.9** (Theorem 4.3 Chappell et al. (2017)). *For any compatible double poset $(P, \preceq_+, \preceq_-)$, there is an explicit homeomorphism between the double chain polytope $\mathcal{C}_2(P)$ and the double order polytope $\mathcal{O}_2(P)$.*

Conveniently, the mapping transfers a triangulation of the double chain polytope to a triangulation of the double order polytope also due to Chappell et al. (2017). This triangulation will be indexed by objects called non-interfering pairs of chains.

**Definition 3.10** (Chappell et al. (2017)). *Given poset $(P, \preceq)$, $I \subset P$ is a* filter *if $x \in I$ and $x \preceq y$ implies $y \in I$, and the* Birkhoff poset $\mathcal{J}(P)$ *is the poset given by the filters of $P$ ordered by inclusion. Given double poset $(P, \preceq_+, \preceq_-)$, for chains of filters $C_+ \subset \mathcal{J}(P_+)$ and $C_- \subset \mathcal{J}(P_-)$, we say the pair of chains $C = (C_+, C_-)$ is* non-interfering *if $\min(J_+) \cap \min(J_-) = \emptyset$ for all $J_+ \in C_+, J_- \in C_-$, where $\min(\cdot)$ denotes the set of minimal elements of the filter. Moreover, we denote by $\Delta^{ni}(P)$ the collection of non-interfering pairs of chains.*

We therefore obtain the following triangulation of the double order polytope indexed by non-interfering pairs of chains. This follows from Corollary 4.1, Theorem 4.3, and Equation 13 of Chappell et al. (2017).

**Lemma 3.11** (Chappell et al. (2017)). *Let $(P, \preceq_+, \preceq_-)$ be a double poset. For $L \subset \mathcal{J}(P)$, define $F(L) = CH(\{\mathbb{1}_J \mid J \in L\}) \subset \mathbb{R}^{|P|}$. For non-interfering pair of chains $C = (C_+, C_-)$ of size $|C| = |C_+| + |C_-| = |P| + 1$, define map $\bar{F}(C) = F(C_+) \boxminus F(C_-)$. Then $\{\bar{F}(C) \mid C \in \Delta^{ni}(P)\}$ is a triangulation of $\mathcal{O}_2(P)$ into $|P|$-dimensional simplices. Moreover, given $C$, $\bar{F}(C)$ can be computed in time $O(d^2)$.*

As a result of this triangulation, it now suffices to uniformly sample a non-interfering pair of chains associated with $P^* - r$ of size $d + 1$. This will be easier, because we can construct a bijection between these pairs and extended bipartitions of $P^* - r$, and extended bipartitions are conceptually simpler than non-interfering pairs of chains.

### 3.3 EXTENDED BIPARTITIONS

**Definition 3.12.** *Given poset $(P, \preceq)$, an* extended bipartition *is a quadruple $((A, \preceq), (B, \preceq), \preceq_A, \preceq_B)$ where $(A, \preceq), (B, \preceq)$ are subposets of $P$ such that $A \cap B = \emptyset$, $A \cup B = P$, and $\preceq_A, \preceq_B$ are linear extensions of $(A, \preceq), (B, \preceq)$ respectively.*

At a high level, given a non-interfering pair of chains, it can be shown that the set difference between the minimal elements of adjacent filters in each chain has size exactly 1, and that these successive differences form a linear extension of the subposet induced by those elements. Moreover, the ground sets of the two linear extensions form a bipartition of the original ground set. Conversely, given an extended bipartition, we can define a non-interfering pair of chains of filters by treating each suffix of each linear extension of the bipartition as the minimal elements of a filter. Due to space constraints, the full proof of this result appears in Appendix A.

**Lemma 3.13.** *There is a bijection between the set of non-interfering pairs of chains $C = (C_+, C_-)$ of the double poset $(P^* - r, \preceq, \preceq)$ of size $|C| = |C_+| + |C_-| = d + 1$ and the set of extended bipartitions of $P^* - r$. Moreover, given an extended bipartition, its corresponding non-interfering pair of chains can be computed in time $O(d^2)$.*

The problem is now reduced to uniformly sampling an extended bipartition. Constructing such a sampler is the last technical step in our argument.

**Lemma 3.14.** *Given poset $(P, \preceq)$ of size $|P| = d$, there is an algorithm to uniformly sample an extended bipartition of $P$ in time $O(d^2)$.*

*Proof.* We start by reformulating the matrix $M$ associated with $P$ (Definition 2.7) as a more suitable data structure $G_P$: iterate through the matrix $M$ and, for each element $v$, construct a list $p_v$ of its parent elements and a list $c_v$ of its child elements. This takes a single $O(d^2)$ pass through $M$. Note that $G_P$ enables us to compute a maximal element in time $O(d)$ by starting at any node and following parent pointers.

Our algorithm, $\mathcal{A}$, receives a $G_P$ representation of a poset $P$ and uses recursion as follows. At each step, compute maximal element $v \in P$ in $O(d)$ and then construct $G_{P-v}$ from $G_P$ by deleting $v$ and the associated parent pointers from its children in time $O(d)$. Then compute $\mathcal{A}(G_{P-v})$ to obtain extended bipartition $((A, \preceq), (B, \preceq), \preceq_A, \preceq_B)$ of $P - v$. We will use this extended bipartition of $P - v$ to construct one for $P$. Note that $\preceq_A$ and $\preceq_B$ can be represented by increasing ordered lists of elements of $A$ and $B$ respectively.

Let $a_j$ be the $\preceq_A$-largest element that is $\preceq$-smaller than $v$ in $a_1 \preceq_A \cdots \preceq_A a_k$. To find $a_j$, iterate from right to left (i.e., $\preceq_A$-largest to $\preceq_A$-smallest) over the $\preceq_A$ list and check whether the current node $w$ has the property that $M_{w,v} = 1$, stopping when the first child of $v$ is found. Define $b_{j'}$ similarly. Then modifying $\preceq_A$ by inserting $v$ anywhere to the right of $a_j$ in $a_1 \preceq \cdots \preceq a_k$ produces a linear extension $\preceq_{A'}$ and the extended bipartition $((A \cup \{v\}, \preceq), (B, \preceq), \preceq_{A'}, \preceq_B)$ of $P$. We call the set of all such insertion points the valid placements for $A$, and define the valid placements for $B$ similarly. Let $L$ be the union of the valid placements for $A$ and $B$. By the above, computing $L$ and uniformly sampling a valid placement takes time $O(d)$. Finally, inserting $v$ into a linear extension takes time $O(d)$ assuming the linear extensions are implemented using linked lists. Overall, we do $O(d)$ work at each recursive step.

In the base case that $|P| = 1$, we flip a coin to either return $((P, \emptyset), \preceq_P, \preceq_\emptyset)$ or $((\emptyset, P), \preceq_\emptyset, \preceq_P)$ where $\preceq_P$ and $\preceq_\emptyset$ are trivial linear extensions. Since we do $O(d)$ work at each recursive step and there are $|P| = d$ steps, the total run time is $O(d^2)$.

It remains to verify that the sampling is uniform. For any extended bipartition of $(P, \preceq)$ given by $((A, \preceq), (B, \preceq), \preceq_A, \preceq_B)$, removing maximal $v$ yields a smaller extended bipartition of the subposet $(P - v, \preceq)$ such that if $a_j$ and $b_{j'}$ are defined with respect to the smaller extended bipartition, then $v$ is either to the right of $a_j$ in $\preceq_A$ of $A$ or to the right of $b_{j'}$ in $\preceq_B$ of $B$. $\qquad\square$

## 3.4 OVERALL ALGORITHM

Having sampled an extended bipartition, Lemma 3.13 already verified that we can convert it to a non-interfering pair of chains efficiently, and Lemma 3.11 shows how to map that to a simplex in the double order polytope in time $O(d^2)$. Sampling the simplex takes time $O(d \log(d))$ (Lemma 3.6), so putting these steps together yields the overall sampler. Pseudocode appears in Algorithm 1.

**Theorem 3.15.** *The poset ball for $(P^*, \preceq)$ can be sampled in time $O(d^2)$.*

---

**Algorithm 1** Poset Ball Sampler

---

1: **Input:** Poset $P^*$ satisfying Assumption 2.8
2: Uniformly sample an extended bipartition $(N_+, N_-, \preceq_A, \preceq_B)$ (Lemma 3.14)
3: Convert $(N_+, N_-, \preceq_A, \preceq_B)$ into its non-interfering chain $C = (C_+, C_-)$ (Lemma 3.13)
4: Compute the vertices of the simplex $\bar{F}(C) = F(C_+) \boxminus F(C_-)$ (Lemma 3.11)
5: Return a uniform sample from $\bar{F}(C)$ (Lemma 3.6)

---

### 3.5 REJECTION SAMPLING THE POSET BALL IS INEFFICIENT

We complement our efficient sampler with a negative result for rejection sampling the poset ball using any $\ell_p$ ball. First, the best possible candidate for rejection sampling is the $\ell_\infty$ ball.

**Lemma 3.16.** *The unit $\ell_\infty$ ball $B_\infty^d$ is the minimum volume $\ell_p$ ball containing the poset ball.*

*Proof.* Any $\ell_p$ ball that contains $(1, .., 1)$ must also contain all points in $\{0,1\}^d - \{(1, ..., 1)\}$ since each of these other points has strictly smaller $\ell_p$ norm. In particular, any $\ell_p$ ball that contains $(1, ..., 1)$ must contain $B_\infty^d$. Since the poset ball contains $(1, \ldots, 1)$, the claim follows. □

Our overall result thus needs only to analyze rejection sampling from $B_\infty^d$.

**Theorem 3.17.** *Rejection sampling the poset ball using any $\ell_p$ ball is inefficient.*

*Proof.* By Lemma 3.16, it suffices to consider $B_\infty^d$. Consider the poset $(P, \preceq)$ on $\{r, p_1, \ldots, p_{d-1}\}$ with set of relations $\{p_i \preceq r\}_{i=1}^{d-1}$. Then for any other poset $P'$, its poset ball has smaller volume as $R(P) \subset R(P')$ means that $P'$ can be formed from $P$ by intersecting $P$ with half-spaces defined by the relations corresponding to the relations in $R(P') - R(P)$. Then $P$'s poset ball is a cylinder with bases $[0,1]^{d-1}$ and height 2 (in the direction of the root axis) so has volume $1^{d-1}(2) = 2$. However, $|B_\infty^d| = 2^d$. It follows that rejection sampling requires $\Omega(2^d)$ samples from $B_\infty^d$ in expectation. □

## 4 EXPERIMENTS

This section evaluates our $K$-norm mechanism on a variety of poset structures . First, we derive a general result about the squared expected $\ell_2$ norm of $\ell_p$ balls (Section 4.1). Based on this result, the strongest comparison for our algorithm is the $\ell_\infty$ mechanism. We use this as the baseline for experiments on path posets, random posets (Section 4.3) and the National Health Interview Survey (Services & Medicaid, 2024) (Section 4.4). An evaluation of runtime appears in Section 4.5.

Note that our mechanism augments a given $d$-element poset with a root to ensure Assumption 2.8 as necessary. Since the baseline $\ell_\infty$ mechanism is applied to the $d$-dimensional problem, we ignore the root dimension of the $(d+1)$-dimensional noise vector produced by our mechanism when comparing squared $\ell_2$ norms. As discussed in Section 2.2, by post-processing, this does not affect privacy.

### 4.1 CHOICE OF BASELINE

This section justifies our choice of the $\ell_\infty$ mechanism — i.e., the $K$-norm mechanism instantiated with the $\ell_\infty$ norm — as the baseline in our experiments. This choice relies on the following result, proved in Appendix B.

**Lemma 4.1.** *Let $\mathbb{E}_2^2(X)$ denote the expected squared $\ell_2$ norm of a uniform sample from $X$, and let $B_p^d$ denote the $d$-dimensional unit $\ell_p$ ball. Then $\mathbb{E}_2^2(B_p^d) = \frac{d}{3}\left(\frac{3d}{d+2}\right)\left(\frac{\Gamma(\frac{d}{p})\Gamma(\frac{3}{p})}{\Gamma(\frac{1}{p})\Gamma(\frac{d+2}{p})}\right)$, where $\Gamma$ is the gamma function, and $\mathbb{E}_2^2(B_\infty^d) = d/3$. Let $r_{p,d} = d^{1/p}$ be the minimum radius for which $r_{p,d}B_p^d$ contains $\mathcal{Y}(P^*)$. By the above, $\mathbb{E}_2^2(r_{p,d}B_p^d) = r_{p,d}^2\mathbb{E}_2^2(B_p^d)$.*

Remark 4.2 from Hardt & Talwar (2010) establishes that, fixing privacy parameters, the expected squared $\ell_2$ norm of a sample from the $K$-norm mechanism is proportional to the expected squared

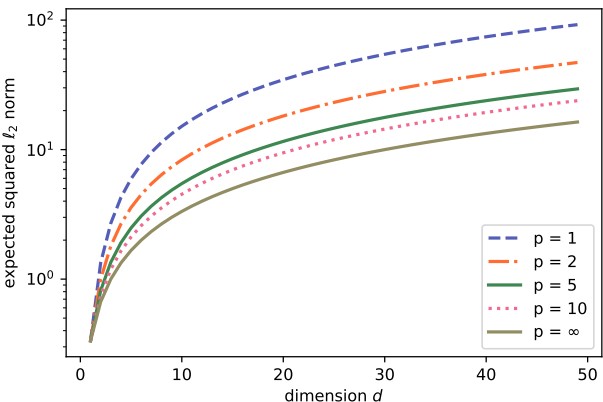

Figure 2: $r_{p,d}^2 \mathbb{E}_2^2(B_p^d)$ (see Lemma 4.1).

$\ell_2$ norm of its norm ball. Thus, to identify the best $\ell_p$-norm mechanism, it suffices to identify the $p$ minimizing $\mathbb{E}_2^2(r_{p,d}B_p^d)$ in Lemma 4.1.

Figure 8 plots the analytic expression derived in Lemma 4.1 for various $p$ and $d$ and leads us to take the $\ell_\infty$-norm mechanism as the strongest baseline.

We note briefly that Laplace ($\ell_1$) noise is approximately 2 ($d = 2$) to 5 ($d = 50$) times worse than $\ell_\infty$ noise by squared $\ell_2$ norm, so the following results are even stronger when evaluated against the Laplace mechanism. Furthermore, for $(1, 10^{-6})$-DP, the analytical Gaussian mechanism (Balle & Wang, 2018) is also dominated by the $\ell_\infty$ mechanism over the range $d \leq 50$ featured in our experiments (see plot in Appendix B.2), so we omit it here.

### 4.2 PATH POSET

We start with the path poset defined by the ground set $P = \{1, ..., d\}$ and the corresponding linear order $1 \preceq 2 \preceq ... \preceq d$ (see Figure 1 for an illustration of its poset ball for $d = 2$). Figure 3 demonstrates that our $K$-norm mechanism improves significantly over the baseline on the path poset, up to over an order of magnitude near $d = 50$. As subsequent experiments will develop, this dramatic error reduction is a consequence of the path graph's extreme depth. Informally, it is the graph on $d$ elements with the smallest poset ball, since any other graph can be created from the path graph by repeatedly moving the leaf element of the original path and grafting it somewhere higher up the tree, and this operation strictly decreases the number of relations in the poset.

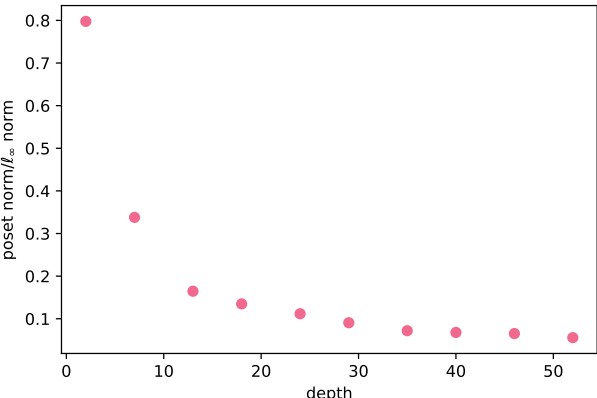

Figure 3: Each point records the average mean squared $\ell_2$ norm ratio between the poset and $\ell_\infty$ balls for the path graph over 100 trials for each of various $d$.

## 4.3 RANDOM POSETS

The second set of experiments features random posets. The posets are generated using the algorithm given by Melançon et al. (2001) for uniformly sampling a directed acyclic graph on a fixed number of uniquely labeled vertices. As their algorithm has runtime $O(d^4)$, we restrict attention to relatively small values for the number of poset elements $d$.

The first random poset experiment examines our algorithm's improvement over the $\ell_\infty$ mechanism as $d$ grows. Figure 4 demonstrates that our algorithm's advantage in terms of expected squared $\ell_2$ norm widens with $d$, increasing to over $90\%$ at $d = 40$.

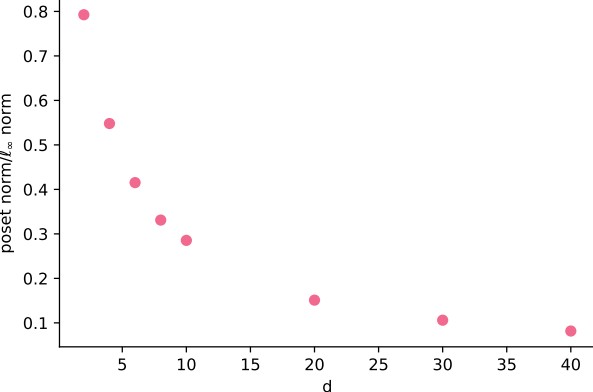

Figure 4: Each point records the average mean squared $\ell_2$ norm ratio between the poset and $\ell_\infty$ balls over 100 trials of random posets. A lower value means our mechanism achieves lower error.

The second and third random poset experiments break this data out along two dimensions (Figure 5). Fixing $d = 10$ (ignoring the root), we plot improvement as a function of both the depth (longest chain of relations) and number of relations in the poset. Both numbers are computed on the transitive reduction of the directed acyclic graph corresponding to the poset, i.e., the directed acyclic graph that minimizes the number of edges while preserving all paths of the poset. As observed for $d$, at a high level we find that larger and richer poset structures lead to correspondingly more dramatic improvements, up to a 4x reduction in error.

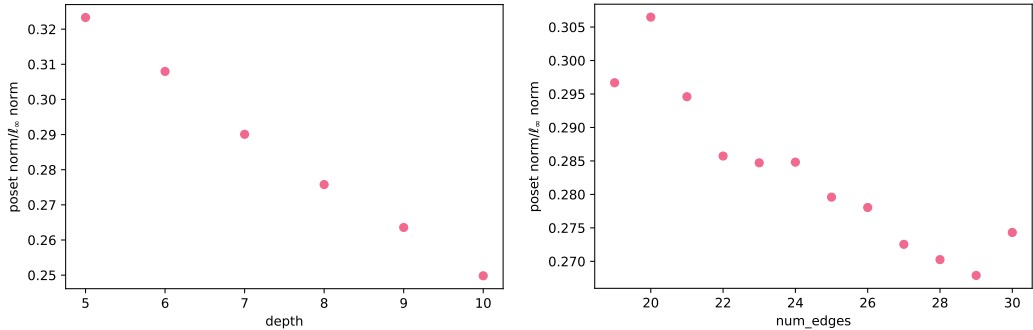

Figure 5: Each point records the average mean squared $\ell_2$ norm ratio between the poset and $\ell_\infty$ balls over at least 100 trials. We omit extremes of both depth and number of edges, as they do not occur enough times in our overall sample of 5000 random posets to cross the 100-sample threshold.

## 4.4 NATIONAL HEALTH INTERVIEW SURVEY

The final set of experiments uses the National Health Interview Survey (NHIS) (Services & Medicaid, 2024). As described in the introduction, the survey includes or omits certain questions depending on previous answers. This induces a poset, and our experiments use either the first, first two,

or first three sections of the survey (Hypertension, Cholesterol, and Asthma). The resulting posets have size from $d = 4$ to $d = 15$. As shown in Figure 6, our mechanism more roughly halves the error of the baseline mechanism.

| # survey sections | poset ball squared $\ell_2$ norm / $l_\infty$ ball squared $\ell_2$ norm |
|---|---|
| 1 | 0.573 |
| 2 | 0.503 |
| 3 | 0.460 |

Figure 6: NHIS average mean squared $\ell_2$ norm ratios, from 10,000 trials each.

## 4.5 RUNTIME

Empirically comparing our algorithm's speed to the general $\tilde{O}(d^{3+\omega})$ sampler from Laddha et al. (2020) is hard, as their algorithm is complex and relies on asymptotic statements that make accurate implementation difficult – to the best of our knowledge, no public implementation of the algorithm exists. However, in addition to its $O(d^2)$ runtime, we provide experiments demonstrating the empirical speed of our algorithm over random graphs of varying depth. In Figure 7 we plot the average runtime of sampling the poset ball as the dimension $d$ varies. On a 2 CPU machine with 32GB RAM, our method takes less than half a second for any of the $d$ used in our experiments.

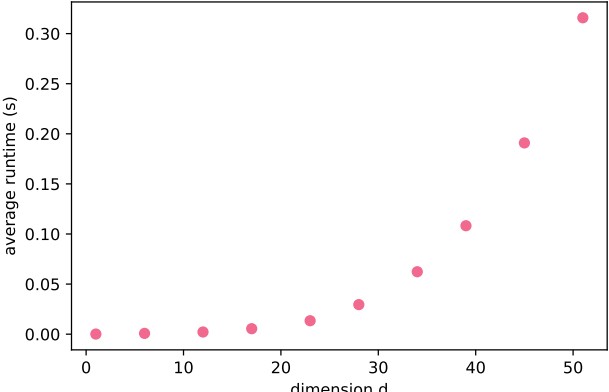

Figure 7: Poset ball sampler average runtime to generate one sample. For each depth we average over 10 different uniformly random posets and over 10 samples of the mechanism.

## 5 DISCUSSION

The material above demonstrates that the $K$-norm mechanism offers strong utility improvements over existing pure differentially private mechanisms for privately counting partially ordered data without sacrificing efficiency. A natural direction for future work is to consider the same problem under approximate differential privacy. There, elliptic Gaussian noise is the primary tool, and the geometric problem changes from sampling the induced norm ball to computing the "smallest" (in terms of expected squared $\ell_2$ norm) ellipse enclosing the induced norm ball. Unfortunately, while Joseph & Yu (2024) solved this problem for count and vote, their solutions relied on the norm balls being symmetric around the vector $(1, \ldots, 1)$, and this does not hold for the poset ball in general. This suggests that new techniques may be required to obtain a similar result for partially ordered data.

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

## A  SAMPLER PROOFS

**Remark A.1.** *The exposition from Chappell et al. (2017) defines the double order polytope of a double poset $(P, \preceq_+, \preceq_-)$ as $\mathcal{O}(2P_+) \boxminus \mathcal{O}(2P_-)$, defines the double chain polytope as $\mathcal{C}(2P_+) \boxminus \mathcal{C}(2P_-)$ and for a pair of chains $C = (C_+, C_-)$ it defines $\bar{F}(C) = 2F(C_+) \boxminus 2F(C_-)$, and $\bar{F}'(C) = 2F'(C_+) \boxminus 2F'(C_-)$. Removing these factors of 2 from the definitions does not change the combinatorial structure of the double order polytope or double chain polytope since this removal simply scales down the $(d-1)$ dimensions of $\mathbb{R}^d$ that excludes the dimension that is added from the $\boxminus$ operation. Consequently, the triangulation of lemma 3.11 still holds. The reason we remove the factors of 2 is because the sensitivity space we are interested in is $CH(\mathcal{O}(P^*) \cup -\mathcal{O}(P^*))$ which is equivalent to our definition of the double order polytope $\mathcal{O}_2(P^* - r)$ where $r$ is the global maximum of $P$.*

**Claim A.2.** *Let $\mathcal{P}$ be a $(d-1)$-dim convex polytope in $\mathbb{R}^d$ lying in the hyperplane $x_1 = 1$. Let $\mathcal{P}' = \mathcal{P} \cup \{0\}$. Then $V(CH(\mathcal{P}' \cup -\mathcal{P}')) = V(\mathcal{P}) \cup V(-\mathcal{P})$.*

*Proof.* We have $V(CH(\mathcal{P}' \cup -\mathcal{P}')) = V(CH(V(\mathcal{P}') \cup V(-\mathcal{P}'))) \subseteq V(\mathcal{P}') \cup V(-\mathcal{P}') = V(\mathcal{P}) \cup V(-\mathcal{P}) \cup \{0\}$ where the $\subseteq$ comes from the fact that the vertices of the convex hull of a finite set is a subset of that set. Since $CH(\mathcal{P}' \cup -\mathcal{P}')$ has antipodal symmetry, then $x \in CH(\mathcal{P}' \cup -\mathcal{P}')$ implies $-x \in CH(\mathcal{P}' \cup -\mathcal{P}')$ which means that the line segment between $x$ and $-x$ contains a small 1-dim neighborhood $B$ around the origin such that $B \subset CH(\mathcal{P}' \cup -\mathcal{P}')$, so the origin is not a vertex of $CH(\mathcal{P}' \cup -\mathcal{P}')$, so we have that $V(CH(\mathcal{P}' \cup -\mathcal{P}')) \subset V(\mathcal{P}) \cup V(-\mathcal{P})$.

Now, suppose that $v \in V(\mathcal{P})$ is not a vertex of $CH(\mathcal{P}' \cup -\mathcal{P}')$. We call a convex combination of vertices non-singleton if its support contains at least 2 vertices. The supposition implies there is a non-singleton convex combination of vertices in $V(\mathcal{P}') \cup V(-\mathcal{P}')$ that equals $v$. None of vertices in the support of this combination can lie in $V(-\mathcal{P}')$ because then the $x_1$ coordinate of the combination would be less than 1 whereas the first coordinate of $v$ is 1. So $v$ must be a non-singleton convex combination of points in $V(\mathcal{P})$, contradicting the fact that $v$ is a vertex of $V(\mathcal{P})$. So $V(\mathcal{P}) \subset V(CH(\mathcal{P}' \cup -\mathcal{P}'))$ and similarly $V(-\mathcal{P}) \subset V(CH(\mathcal{P}' \cup -\mathcal{P}'))$. But by the previous paragraph, $V(CH(\mathcal{P}' \cup -\mathcal{P}')) \subset V(\mathcal{P}) \cup V(-\mathcal{P})$, so $V(CH(\mathcal{P}' \cup -\mathcal{P}')) = V(\mathcal{P}) \cup V(-\mathcal{P})$. $\square$

**Lemma 3.13.** *There is a bijection between the set of non-interfering pairs of chains $C = (C_+, C_-)$ of the double poset $(P^* - r, \preceq, \preceq)$ of size $|C| = |C_+| + |C_-| = d + 1$ and the set of extended bipartitions of $P^* - r$. Moreover, given an extended bipartition, its corresponding non-interfering pair of chains can be computed in time $O(d^2)$.*

*Proof.* Suppose we have a non-interfering pair of chains $C = (C_+, C_-)$ where $|C| = d + 1$. Write each chain of filters as $C_\sigma = J_1^\sigma \subsetneq ... \subsetneq J_{|C_\sigma|}^\sigma$ for $\sigma \in \{+, -\}$. Then $|\min(J_{i+1}^\sigma) - \min(J_i^\sigma)| \geq 1$ for $1 \leq i \leq |C_\sigma| - 1$. Any element $p \in (\min(J_{i+1}^\sigma) - \min(J_i^\sigma))$ lies in $J_{i+1}^\sigma$. If it also lies in $J_i^\sigma$, then $p \notin \min(J_i^\sigma)$ means there exists $p' \in J_i^\sigma$ with $p' \preceq p$, but $J_i^\sigma \subsetneq J_{i+1}^\sigma$ then implies that $p \notin \min(J_{i+1}^\sigma)$, a contradiction. Thus $(\min(J_{i+1}^\sigma) - \min(J_i^\sigma)) \subseteq (J_{i+1}^\sigma - J_i^\sigma)$, and similarly for any $j > i$, we have

$$(\min(J_{j+1}^\sigma) - \min(J_j^\sigma)) \subseteq (J_{j+1}^\sigma - J_j^\sigma).$$

Then $J_{i+1}^\sigma \subseteq J_j^\sigma$ implies $(\min(J_{i+1}^\sigma - \min(J_i^\sigma)) \cap (\min(J_{j+1}^\sigma - \min(J_j^\sigma)) = \emptyset$.

Let $N_\sigma = \cup_{i=1}^{|C_\sigma|-1}(\min(J_{i+1}^\sigma) - \min(J_i^\sigma))$. The above implies $|N_\sigma| \geq |C_\sigma| - 1$. Moreover, $N_+ \cap N_- = \emptyset$ since $N_\sigma \subset \left(\cup_{i=1}^{|C_\sigma|-1} \min(J_{i+1}^\sigma)\right)$, and $\left(\cup_{i=1}^{|C_+|-1} \min(J_{i+1}^+)\right) \cap \left(\cup_{i=1}^{|C_-|-1} \min(J_{i+1}^-)\right) = \emptyset$ since $C$ is non-interfering. Thus

$$|N_+ \cup N_-| = |N_+| + |N_-| \geq |C_+| + |C_-| - 2 = |C| - 2 = d - 1 = |P^* - r|.$$

But $N_+ \cup N_- \subseteq P^* - r$, so $|N_+ \cup N_-| = |P^* - r|$. This means that $(N_+, N_-)$ partition $P^* - r$ and $|\min(J_{i+1}^\sigma) - \min(J_i^\sigma)| = 1$ for $1 \leq i \leq |C_\sigma| - 1$. It follows that $\min(J_1^\sigma) = J_1^\sigma = \emptyset$. This leads us to the following extended bipartition. Let $a_i = \min(J_{|C_+|-i+1}^+) - \min(J_{|C_+|-i}^+)$ for $1 \leq i \leq |C_+| - 1$ and $b_i = \min(J_{|C_-|-i+1}^-) - \min(J_{|C_-|-i}^-)$ for $1 \leq i \leq |C_-| - 1$. Since filters are upwards closed, either $a_i \preceq a_j$ for some $a_j \in \{a_{i+1}, ..., a_{|C_+|-1}\}$ or $a_i$ is incomparable to each element of $\{a_{i+1}, ..., a_{|C_+|-1}\}$. In either case, $a_1 \preceq_A ... \preceq_A a_{|C_+|-1}$ is a linear extension of $N_+$, and similarly $b_1 \preceq_B ... \preceq_B b_{|C_-|-1}$ is a linear extension of $N_-$, so $((N_+, \preceq), (N_-, \preceq), \preceq_A, \preceq_B)$ is an extended bipartition of $P^* - r$.

Conversely, given an extended bipartition $(N_+, N_-, a_1 \preceq_A ... \preceq_A a_{|N_+|}, b_1 \preceq_B ... \preceq_B b_{|N_-|})$, define sets $m_1^+ = \emptyset$, $m_1^- = \emptyset$, $m_i^+ = \min(\cup_{j=1}^{i-1} a_{|N_+|+1-j})$ for $2 \leq i \leq |N_+| + 1$ and $m_i^- = \min(\cup_{j=1}^{i-1} b_{|N_-|+1-j})$ for $2 \leq i \leq |N_-| + 1$. Define $J_i^\sigma$ to be the filter with minimal elements $m_i^\sigma$ for $2 \leq i \leq |N_\sigma| + 1$ and let $J_1^\sigma = \emptyset$. Let $C_\sigma = J_1^\sigma \subsetneq ... \subsetneq J_{|N_\sigma|+1}^\sigma$. Then $(C_+, C_-)$ is a non-interfering pair with $|C_+| + |C_-| = |N_+| + |N_-| + 2 = d + 1$, and this map from the set of extended bipartitions to the set of non-interfering pairs of chains is an inverse of the mapping of the previous paragraphs.

It remains to verify the runtime of computing the non-interfering pair of chains we constructed above. Without loss of generality, we focus on $J_i^+$. Filter $J_1^+ = \emptyset$ and $J_2^+$ is the set of elements with

a 1 in row $|N_+|$ of matrix $M$, computed in time $O(d)$. Now, suppose we have computed $J_i^+$ and want to compute $J_{i+1}$. By comparing rows $|N_+| + 1 - i$ and $|N_+| + 2 - i$ of $M$, we can compute the set $p_i$ of parents of $a_{|N_+|+1-i}$ that are not parents of $a_{|N_+|+2-i}$ (we include $a_{|N_+|+1-i} \in p_i$) in time $O(d)$. Note that $p_i = J_{i+1} - J_i$. Then $J_{i+1}^+ = J_i^+ \sqcup p_i$ where $\sqcup$ is disjoint union. Thus each filter of $J_1^+ \subsetneq \ldots \subsetneq J_{|N_+|+1}^+$ can be computed in time $O(d)$, taking time $O(d^2)$ overall. $\qquad\square$

## B  EXPERIMENT PROOFS

### B.1  EXPECTED SQUARED NORM

We start with a simple result about the expected squared $\ell_2$ norm, $\mathbb{E}_2^2$. Recall from Section 4.1 that $\mathbb{E}_2^2(X)$ denotes the expected squared $\ell_2$ norm of a uniform sample from $X$.

**Lemma B.1.** *Then $\mathbb{E}_2^2(X \oplus Y) = \mathbb{E}_2^2(X) + \mathbb{E}_2^2(Y)$.*

*Proof.* To uniformly sample $X \oplus Y$, we can uniformly sample $x \in X, y \in Y$ and then return $x \oplus y$. Samples $x$ and $y$ embed into disjoint coordinates of $x \oplus y$, so $\|x \oplus y\|_2^2 = \|x\|_2^2 + \|y\|_2^2$. $\qquad\square$

**Lemma 4.1.** *Let $\mathbb{E}_2^2(X)$ denote the expected squared $\ell_2$ norm of a uniform sample from $X$, and let $B_p^d$ denote the $d$-dimensional unit $\ell_p$ ball. Then $\mathbb{E}_2^2(B_p^d) = \frac{d}{3}\left(\frac{3d}{d+2}\right)\left(\frac{\Gamma(\frac{d}{p})\Gamma(\frac{3}{p})}{\Gamma(\frac{1}{p})\Gamma(\frac{d+2}{p})}\right)$, where $\Gamma$ is the gamma function, and $\mathbb{E}_2^2(B_\infty^d) = d/3$. Let $r_{p,d} = d^{1/p}$ be the minimum radius for which $r_{p,d}B_p^d$ contains $\mathcal{Y}(P^*)$. By the above, $\mathbb{E}_2^2(r_{p,d}B_p^d) = r_{p,d}^2 \mathbb{E}_2^2(B_p^d)$.*

*Proof.* We use the fact that $\int_0^1 t^{a-1}(1-t)^{b-1} = \frac{\Gamma(a)\Gamma(b)}{\Gamma(a+b)}$. Then

$$
\begin{aligned}
\mathbb{E}_2^2(B_p^d) &= |B_p^d|^{-1} \int_{|x_1|^p+\ldots+|x_d|^p \leq 1} \sum_{i=1}^d x_i^2 \partial x_1 \ldots \partial x_d \\
&= |B_p^d|^{-1} \int_{|x_1|^p+\ldots+|x_d|^p \leq 1} d x_1^2 \partial x_1 \ldots \partial x_d \\
&= |B_p^d|^{-1} \int_{-1}^1 d x_1^2 \partial x_1 \int_{|x_2|^p+\ldots+|x_d|^p \leq 1-|x_1|^p} \partial x_2 \ldots \partial x_d \\
&= |B_p^d|^{-1} \int_{-1}^1 d x_1^2 |B_p^{d-1}| (1-|x_1|^p)^{\frac{d-1}{p}} \partial x_1 \\
&= 2d|B_p^d|^{-1}|B_p^{d-1}| \int_0^1 x^2 (1-x^p)^{\frac{d-1}{p}} \partial x.
\end{aligned}
$$

We substitute $y = x^p$ to get

$$
\begin{aligned}
\partial y &= px^{p-1}\partial x \\
\partial x &= \frac{1}{px^{p-1}}\partial y \\
\partial x &= \left(\frac{1}{p}\right) y^{\frac{1}{p}-1}\partial y
\end{aligned}
$$

and the chain of equalities continues with

$$
\begin{aligned}
\mathbb{E}_2^2(B_p^d) &= \left(\frac{2}{3}\right) d|B_p^d|^{-1}|B_p^{d-1}| \int_0^1 y^{\frac{2}{p}}(1-y)^{\frac{d-1}{p}} \left(\frac{3}{p}\right) y^{\frac{1}{p}-1} \partial y \\
&= \left(\frac{2}{3}\right) d|B_p^d|^{-1}|B_p^{d-1}| \left(\frac{3}{p}\right) \int_0^1 y^{\frac{3}{p}-1}(1-y)^{\frac{d-1}{p}} \partial y \\
&= \left(\frac{2}{3}\right) d|B_p^d|^{-1}|B_p^{d-1}| \left(\frac{3}{p}\right) \left(\frac{\Gamma\left(\frac{3}{p}\right)\Gamma(\frac{d-1}{p}+1)}{\Gamma(\frac{d+2}{p}+1)}\right) \\
&= \left(\frac{2}{3}\right) d \left(\frac{(2\Gamma(\frac{1}{p}+1))^{d-1}}{\Gamma(\frac{d-1}{p}+1)}\right) \left(\frac{\Gamma(\frac{d}{p}+1)}{(2\Gamma(\frac{1}{p}+1))^d}\right) \left(\frac{\Gamma(\frac{3}{p}+1)\Gamma(\frac{d-1}{p}+1)}{\Gamma(\frac{d+2}{p}+1)}\right) \\
&= \left(\frac{2}{3}\right) d \left(\frac{\Gamma(\frac{d}{p}+1)}{2\Gamma(\frac{1}{p}+1)}\right) \left(\frac{\Gamma(\frac{3}{p}+1)}{\Gamma(\frac{d+2}{p}+1)}\right) \\
&= \frac{d}{3} \left(\frac{3d}{d+2}\right) \left(\frac{\Gamma(\frac{d}{p})\Gamma\left(\frac{3}{p}\right)}{\Gamma(\frac{1}{p})\Gamma(\frac{d+2}{p})}\right).
\end{aligned}
$$

For $p = \infty$, we use

$$
\mathbb{E}_2^2(B_\infty^d) = \mathbb{E}_2^2([0,1]^d) = d\mathbb{E}_2^2([0,1]) = d \int_0^1 t^2 dt = \frac{d}{3}
$$

where the first equality comes from the fact that the expectation restricted to the positive orthant does not change the expectation value because of symmetry over the orthants, and the second equality comes from Lemma B.1.

We show that $r_{p,d} = d^{1/p}$. Note that $r_{p,d}$ is equal to the smallest radius such that $r_{p,d}B_p^d$ contains the point $(1, ..., 1) \in \mathcal{Y}(P^*)$ because if $(1, ..., 1) \in r_{p,d}B_p^d$ then all other binary vectors of length $d$ are also contained in $r_{p,d}B_p^d$, i.e. $\mathcal{Y}(P^*) \subset r_{p,d}B_p^d$.

Next, we show that $\mathbb{E}_2^2(r_{p,d}B_p^d) = r_{p,d}^2 \mathbb{E}_2^2(B_p^d)$. Since $r_{p,d}B_p^d = \{r_{p,d}x : x \in B_p^d\}$,

$$
\begin{aligned}
\mathbb{E}_2^2(r_{p,d}B_p^d) &= |B_p^d|^{-1} \int_{|x_1|^p+...+|x_d|^p \leq 1} \sum_{i=1}^d (r_{p,d}x_i)^2 \partial x_1...\partial x_d \\
&= r_{p,d}^2 |B_p^d|^{-1} \int_{|x_1|^p+...+|x_d|^p \leq 1} \sum_{i=1}^d x_i^2 \partial x_1...\partial x_d \\
&= r_{p,d}^2 \mathbb{E}_2^2(B_p^d).
\end{aligned}
$$

$\square$

## B.2 ANALYTICAL GAUSSIAN MECHANISM COMPARISON

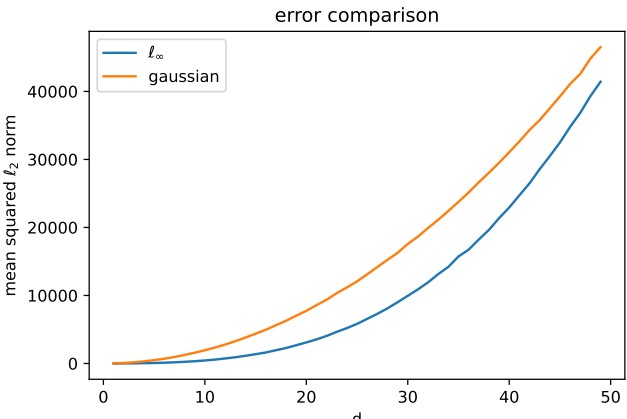

Figure 8: A comparison of expected squared $\ell_2$ norms for the 1-DP $\ell_\infty$ mechanism (with $\ell_\infty$ sensitivity 1) and $(1, 10^{-6})$-DP analytical Gaussian mechanism (with $\ell_2$ sensitivity $\sqrt{d}$).

