# OpenReview forum: "Privately Counting Partially Ordered Data"
_ICLR.cc/2025/Conference — ICLR 2025 Poster_

### Official Review · Reviewer_dkVy · 2024-10-24

**Soundness:** 2
**Presentation:** 3
**Contribution:** 2
**Rating:** 5
**Confidence:** 3

**Summary:**

The paper primarily focuses on differentially private counting when data points satisfying a partial order. Previous best approaches takes time $\tilde{O}(d^{2+\omega})$ where $\omega\geq 2$ is the matrix multiplication exponent. In contrast, this work provides a fast sampler for the induced norm ball that runs in $O(d^2)$. To achieve this, this work starts with connecting the poset ball to double order polytope, then reduces the problem to uniformly sampling a non-interfering pair of chains with the results from Chappell and finally to sampling an extended bipartition. Experiments are conducted to show that the final sampler reduces error over existing DP mechanisms. To complement the sampler, this work also shows a negative result for rejection sampling.

**Strengths:**

S1: This paper addresses the problem by providing a sampler with more than cubic speedup over the previous algorithms.

S2: The proof path leading to the final sampler is clear and well-organized.

S2: This paper also investigates the rejection sampling and provides a negative result.

S3: The paper conducts experiments on both synthetic and real-world partial orders to demonstrate error reductions of their sampler.

**Weaknesses:**

W1: The comparison with previous research, particularly with the work of Joseph & Yu, does not seem sufficiently thorough. (summarized in the questions)

W3: The results of the experiments appear to be incomplete (also summarized in questions.)

**Questions:**

Q1: Although the authors have introduced the differences between their work and that of Joseph & Yu in the related work section, they do not seem to have adequately demonstrated the innovative aspects of their work. Furthermore, the structure of the paper appears similar, as Joseph & Yu also considered rejection sampling. It seems that this paper merely modifies the sampling technique based on changes in the characteristics of the input data, without showcasing its unique core technology.

It would be beneficial if the authors can provide a more detailed comparison, such as explaining how the data considered in this paper differs from that considered by Joseph & Yu; why their sampling technique is not well-suited to our current data; and how the sampling method proposed in this paper leverages the new data characteristics and what innovations it introduces.


Q2: The experiments in the paper only investigate the error reduction of the sampler. Since the main contribution of the paper is a sampler faster then previous algorithms, it would be beneficial for the authors to conduct experiments that demonstrate speedup of their algorithm and compare it with former approaches.

---

> ### Author Response · Authors · 2024-11-18
> **Response to Reviewer dkVy**
>
> 1) We have expanded the discussion of the Joseph and Yu paper in the Related Work section (see updated submission). To summarize, in both papers, the primary technical contribution is developing a fast sampler for the norm ball induced by the sensitivity space. Since the papers consider different problems, these lead to different norm balls. The original idea for sampling such balls to run the K-norm mechanism was introduced by HT10 (https://arxiv.org/abs/0907.3754), but it is impractically slow without these fast and problem-specific samplers. Therefore, while both our paper and Joseph and Yu develop fast problem-specific samplers, the techniques required are quite different, because the norm balls are completely different geometric objects and each sampler has a structure that is highly specific to the object being sampled. For example, their sum polytope triangulation is indexed by permutations with a fixed number of ascents, while ours is indexed by pairs of non-interfering chains in the Birkhoff lattice, and developing uniform samplers for these different objects requires different approaches.
>
> 2) It is hard to provide an empirical comparison to the general $\tilde O(d^{3+\omega})$ sampler given by Laddha et al. (https://arxiv.org/abs/1911.05656), as their algorithm is quite complicated and relies on asymptotic statements that make accurate implementation difficult – to the best of our knowledge, no public implementation of the algorithm exists. However, we have added empirical runtimes for our algorithm (Section 4.5) demonstrating that our sampler runs in under half a second on a typical personal computer for problems at the scale considered in our experiments. We hope that the combination of fast empirical and theoretical runtimes for our algorithm and slow theoretical runtime for the general sampler are sufficient evidence of the speed improvement.

---

> > ### Comment · Reviewer_dkVy · 2024-11-26
> >
> > Thank you for your response. My assessment remains the same.

---

### Official Review · Reviewer_jiCC · 2024-11-04

**Soundness:** 3
**Presentation:** 2
**Contribution:** 3
**Rating:** 8
**Confidence:** 4

**Summary:**

Differentially private algorithms for counting queries have a long history and mechanisms that add carefully chosen noise when queries have some structure can improve accuracy. Some of these algorithms are often used in practice, e.g. by the US Census Bureau. For the case of pure DP, the K-norm mechanism of Hardt and Talwar improves on the Laplace mechanism that does not exploit the structure of the queries. The algorithm however requires sampling from the “sensitivity polytope” defined by the queries.

The current paper studies this sampling problem for the case when the queries have a “partially ordered” structure. E.g. surveys often have a sequence of questions where a No answer to a question implies a no answer to the next question. This imposes a constraint that the answer to $Q_{i+1}$ can be 1 only if $Q_{i}$ has answer 1. The authors study the more general case of partial orders.

The main contribution of this work is to show an efficient algorithm to sample from the sensitivity polytope defined by a partial order. For the case of d queries, the proposed algorithm runs in time $O(d^2)$. The paper follows a general schema in recent work by Joseph and Yu, where the sensitivity polytope is expressed as a union of simplexes. A simplex is easy to sample from, and thus the problem reduces to sampling a random simplex proportional to its volume. While Joseph and Yu used this approach for some set of problems, their work does not cover this natural set of constraints, and this is addressed in the current work.

The main contribution in my mind is to bring in the technical tools from recent works in the geometry of the poset polytopes, and apply them to this natural problem. The authors also do experiments comparing their algorithm to using K-norm for an \ell_p norm, and show that for random posets, using the K-norm improves noticeably the error in the estimate for random posets, as well as for an National Health Interview Survey.

**Strengths:**

- Brings in tools from another research area to improve algorithms/bounds for a natural and practical problem

**Weaknesses:**

- The paper is difficult to read for someone who is not an expert in partial orders.

**Questions:**

- The recursive K-norm mechanism can improve the error bounds. It would be valuable to see the error comparison. Relatedly, Hardt and Talwar seem to have lower bounds and one can also minimize over matrix mechanisms. It would be valuable to compute the lower bounds and see how far off they are from the upper bounds here.
- For larger d, I would expect that (eps, delta)-DP algorithms with a small delta would become competitive quickly, and have the advantage of being easier to sample from. A comparison even with the simplest Gaussian mechanism, for a small delta (say 10^-6 - 10^-9) would make the paper better.
- In the case of surveys, the poset structure would be further restricted to a “product structure”, where Q2-Q5 may depend on Q1, and Q7-Q9 may depend on Q6, but Q1-Q5 and Q6-Q10 are independent of each other. In this case, you should be able to reduce the run time from d^2 to dk, where k is the size of the largest dependency. This seems like a natural extension of the current work and should be written down.

---

> ### Author Response · Authors · 2024-11-18
> **Response to Reviewer jiCC**
>
> 1) Please see the updated submission for a) language clarifying our contributions and the relationship between our work and Chappell et al. (see response to Reviewer a51u for details), and b) a concrete example based on the path poset (see response to Reviewer dNFD for details). We hope this will make the algorithm easier to understand.
>
> 2) The chief weakness of the recursive K-norm mechanism given by HT10 is that it runs the K-norm mechanism on successively smaller subspaces by approximating their covariance matrices. Our efficient sampler still takes $O(d^2)$ time to sample the poset ball once, and their algorithm requires approximately $O(d^4)$ samples to approximate the relevant covariance matrices in each subspace. This blows up the runtime enough to be impractical. If desired, we can add a short discussion of this issue to the Related Work section -- we omitted it in our paper, as Joseph and Yu (https://arxiv.org/pdf/2309.15790, Section 1.2) makes the same point. Regarding lower bounds, we are not sure how to compare the asymptotic lower bounds given by HT10 to the empirical upper bounds provided here – do you have something specific in mind?
>
> 3) For $(1, 10^{-6})$-DP, the Gaussian mechanism is dominated by the $\ell_\infty$ mechanism for the $d \leq 50$ used in our experiments. We have added a short note discussing this in the revised submission (see the end of Section 4.1 and Appendix B.2).
>
> 4) The example provided is a forest with two disconnected components. However, the poset ball of a forest does not in general decompose as a direct sum of the poset balls of its connected components. To see this, let $S(G) = V(O(G)) \cup -V(O(G))$ be the vertices of the poset ball of graph $G$, and let $CH(Z)$ denote the convex hull of points in a set $Z$. Then $CH(S(X \cup Y)) \neq CH(S(X)) \oplus CH(S(Y))$, where $\oplus$ is the direct sum operation. To see this, let $X$ and $Y$ be two different single vertex graphs. Then $CH(S(X)) \oplus CH(S(Y))$ is $[-1,1] \oplus [-1,1]$ which is the $\ell_{\infty}$ unit ball. On the other hand, $CH(S(X \cup Y))$ is the convex hull of the unit square in the positive quadrant and its negation. Consequently, sampling independently from each component and combining the samples via direct sum would not yield a uniform sample from the poset ball.

---

> > ### Comment · Reviewer_jiCC · 2024-11-21
> > **Thanks**
> >
> > Thanks for the responses.
> >
> > K-norm mechanism complexity: While I appreciate the fact that the recursive K norm is quite inefficient to sample from, it would be valuable to understand how much one pays for being efficient. Would it be feasible to run it for d=4? for d=8? Note that recursive K-norm can work with any decomposition into subspaces, and if you have intuition for which directions are the long ones (perhaps the all 1s direction), you can get a heuristic recursive K-norm that may improve on the K-norm.
> >
> > The Gaussian comparison is very useful, and it is interesting to see that even though Gaussian squared error grows as $d^2$, and the linfty mechanism has error growing as $d^3$, the constant factors are significant enough to make the linfty better all the way to $d=50$. This suggests that K-norm may be better than the geometry-tailored Gaussian mechanism for a large range of parameters. It may be quite valuable to do that comparison as well. E.g. is it feasible to compute or estimate the best factorization mechanism for your K, at least for small d? For example the path poset is a convex hull of a small number of points, and it should easy to understand (or numerically compute, using the approach of Li et al.) the right Gaussian noise to add. If so, it is likely to show that your mechanism is better than the best factorization mechanism for small d, and that would make the result even more compelling.

---

> > > ### Author Response · Authors · 2024-11-22
> > >
> > > Thanks for following up!
> > >
> > > > K-norm mechanism complexity...
> > >
> > > Running the recursive $K$-norm mechanism only for small $d$ would mitigate time issues, but implementation challenges remain. First, heuristically determining the subspace decomposition is difficult, as we would need to repeat this instance-specific choice for each of the (hundreds of) posets used in our experiments. We could instead avoid this choice by employing the general covariance matrix estimation approach outlined by HT10 (at the cost of much higher runtime for the estimation), but then it still remains to sample the lower subspaces, for which a general sampler becomes necessary.  Moreover, it is not obvious how to run a general sampler on a projected subspace, as we would also need to compute the set of constraints defining the projected object for each projection step and each problem instance. Finally, due to runtime considerations, these experiments would still only extend to a dimension that covers a minority of the instances we currently consider. For these reasons, we suggest that comparing to the recursive $K$-norm mechanism is not feasible.
> > >
> > > > The Gaussian comparison ...
> > >
> > > It might help to clarify a couple of similar but distinct baselines. An approach discussed in NT24 (https://arxiv.org/abs/2301.13850) is essentially equivalent to finding the minimum (in terms of expected squared $\ell_2$ norm) ellipse enclosing the sensitivity space, and adding Gaussian noise with covariance scaled according to that ellipse. Finding this ellipse is not simple in general -- it is one of the main contributions of JY24 (https://arxiv.org/pdf/2309.15790) -- and, as discussed in Section 5 of the submission, it is unclear how to do this analytically for poset balls. The general approach for this problem is to use a semidefinite program, but $\text{poly}(d)$ convergence is only known for certain (somewhat cryptically defined) classes of problems, and the relevant polynomial has high degree even for these classes (see the discussion at the end of Section 2.2). In the case of factorization mechanisms, a similar optimization is performed, but instead of working directly with the $d$-dimensional sensitivity polytope, databases are represented as histograms over the data universe, with one bin for each possible data point, and neighboring databases are required to be within $\ell_1$ or $\ell_2$ distance 1. For posets, the relevant histograms have size $O(2^d)$. This increases the computational complexity of finding the optimal factorization to be infeasible for most of our experiment instances.
> > >
> > > Finally, we note that, to the best of our knowledge, there are no public implementations of the recursive $K$-norm, general elliptic Gaussian, or factorization mechanisms.

---

> > > > ### Comment · Reviewer_jiCC · 2024-11-30
> > > > **Thanks**
> > > >
> > > > Thanks for your responses. I will be increasing my score.

---

### Official Review · Reviewer_a51u · 2024-11-05

**Soundness:** 3
**Presentation:** 1
**Contribution:** 3
**Rating:** 6
**Confidence:** 2

**Summary:**

The paper presents a sampling algorithm that samples from a K-norm mechanism on an induced ball for counting partially ordered data. The algorithm presented in the paper runs in $O(d^2)$ time instead of $O(d^{2+\omega})$ time.

**Strengths:**

The paper proposes a sampling mechanism for a special instantiation of K-norm mechanism that improves the state of the art sampling algorithm by a factor of $O(d^\omega)$.

**Weaknesses:**

I found the paper rather hard to read and also difficult to figure out what are the contributions of the authors and what follows more or less from Chappell et al. (2017). I suggest that the authors make it explicitly clear by giving a high level overview of their proof stating clearly what steps requires their proof and what was already known in the literature. To me, it feels like the main contribution is the proof of Lemma 3.14, but I can be wrong and would love to stand corrected.

If the authors can give me a better understanding of which aspect of their paper is new, I would be more than happy to increase the score.

**Questions:**

Please give us a good overview of which is your contribution and what are the challenges faced when using the previous algorithms to perform the sampling.

Also, what is the best known sampling algorithm that runs in time $O(d^{2+\omega})$? What happens if that sampling algorithm is instantiated with the set up that the authors propose? What difference from that sampling algorithm do the authors take?

---

> ### Author Response · Authors · 2024-11-18
> **Response to Reviewer a51u**
>
> 1. We have attempted to add a high-level overview of the relationship between our work and Chappell et al. at the beginning of Section 3. We have also added more subsection headings to further clarify our proof structure. Briefly, our theoretical contributions are a) observing the connection between the privacy problem of counting partially ordered data and the combinatorics work of Chappell et al. and translating between the two settings (specifically, we show that sampling from the k-norm mechanism is equivalent to sampling from a double order polytope (Lemma 3.4)), b) proving a bijection between the non-interfering chains of Chappell et al. and extended bipartitions (Lemma 3.13), and c) constructing a fast uniform sampler for the relevant extended bipartitions (Lemma 3.14). Note in particular that extended bipartitions are objects original to our paper, and while Chappell et al. derive a triangulation of the double order polytope, they do not discuss sampling any of those objects.
>
> 2. The best alternative sampler is due to Laddha et al. (https://arxiv.org/abs/1911.05656), and a short discussion appears at the end of Section 2.2. Note that this sampler takes time $\tilde O(d^{3+\omega})$ where $\omega > 2$. The primary operational difference between their sampler and ours is that their sampler can be applied to any polytope; in contrast, our sampler is tailored only to the poset ball, and relies on knowledge of its structure to sample quickly. Note also that applying their sampler only achieves $O(\varepsilon)$-DP, as it is an approximate sampler, whereas ours is exact.

---

> > ### Comment · Reviewer_a51u · 2024-12-01
> > **Post rebuttal**
> >
> > Thanks for the clarification. I have increased my score.

---

### Official Review · Reviewer_dNFD · 2024-11-09

**Soundness:** 3
**Presentation:** 3
**Contribution:** 3
**Rating:** 8
**Confidence:** 3

**Summary:**

This work studies differentially private summation of items that satisfy a partial order. The authors show that the problem can be solved in time $d^2$ where $d$ is the number of bits and outperforms existing private algorithms.

**Strengths:**

1. Summing over items with partial orders is a fundamental problem that has not been fully studied under differential privacy.
2. The proposed algorithm is very time-efficient, only quadratic in the number of bits. The speed-up is significant over some simple sampling algorithms
3. The estimation error is much better than standard privacy algorithms based on $\ell_{\infty}$ norm.

**Weaknesses:**

1. The algorithm is only for pure differential privacy. Approximate DP is sometimes more practical in many applications.
2. It would be helpful to the readers to provide a high-level overview of the algorithm and why it improves over standard algorithms through a simple example.

**Questions:**

How can the algorithm be adapted to approximate DP and achieve better results than pure DP?

---

> ### Author Response · Authors · 2024-11-18
> **Response to Reviewer dNFD**
>
> 1) We agree that extending this approach to approximate DP is a natural next problem. We attempted to discuss this problem in the Discussion section at the end of the paper. Recapping the brief discussion there, to extend to approximate DP, it suffices to find the minimum ellipse (in terms of expected squared $\ell_2$ norm) that contains the poset ball, as done by Joseph and Yu (https://arxiv.org/abs/2309.15790). Finding the minimum enclosing ellipse for general high-dimensional polytopes is a difficult optimization problem, and the general approach relies on semidefinite programs. This approach can be very slow (for details, see the discussion at the end of Section 2.2 in Joseph and Yu). Joseph and Yu circumvent this difficulty by exploiting the symmetry of sum, count, and vote polytopes around the vector $(1,...,1)$, a property not all poset balls possess. Consider, for example, the simple poset with elements $v_1$ and $v_2$, where $v_1 > v_2$: the vector $(1,0)$ is in its ball $P$, while $(0,1)$ is not. This breaks their approach to finding the minimum enclosing ellipse, and it is not clear how to fix this issue.
>
> 2) Several reviewers asked for a simpler high-level overview of the algorithm. We have added an informal discussion of the proof at the beginning of Section 3, as well as an example and experiments based on the path poset ($x_d \leq x_{d-1} \leq \cdots \leq x_1$), as the resulting poset ball is both geometrically simple (it is a linear transformation of the $\ell_1$ ball) and yields a large utility improvement over the $\ell_\infty$ mechanism (see Figure 1 and Section 4.2 in the revised submission).

---

> > ### Comment · Reviewer_dNFD · 2024-11-26
> >
> > Thanks for your response. My evaluation remains the same.

---

### Author Response · Authors · 2024-11-18
**Overall response**

Thanks for the reviews! We have uploaded a new submission with additions highlighted in red. We make one overall note and then provide individual reviewer responses below.

While preparing the response, we noticed an off-by-one error in our experiments. For some experiments (the random posets and NHIS data), the poset does not have a root, so we add one, run the K-norm mechanism on the $(d+1)$-dim data, and then output the last $d$ dimensions of the final output (i.e., drop the added root dimension) as discussed at the beginning of Section 4 in both the original and revised paper. However, our initial submission experiments did so using a radius drawn from a $\Gamma(d+1, 1/\varepsilon)$ distribution, while the correct distribution is $\Gamma(d+2, 1/\varepsilon)$. The new submission corrects this, and the result is a slightly reduced improvement. Note that the change is small – the most dramatic change occurs for the NHIS data, where the average improvement drops from 60% to 50%. Nonetheless, we highlight this change for transparency.

---

### Meta-Review · Area_Chair_tn93 · 2024-12-22

**Metareview:**

The paper studies the problem of designing differentially private algorithms for counting queries that have a partially ordered structure. The main contribution of the paper is an algorithm guaranteeing pure differential privacy while achieving a running time that is significantly faster than prior work.

Designing differentially private algorithms for counting queries is an important direction with many applications. The direction of exploiting additional structure is a well motivated one. The algorithm proposed is very efficient and practical, and the experimental evaluation shows that the proposed algorithm achieves significant improvements in the error. Most of the reviewers appreciated the main contribution and showed strong support for the paper. One of the reviewers was concerned that the paper is not sufficiently novel compared to the prior work of Joseph and Yu. The author response addressed this concern and noted that the techniques are quite different.

**Additional Comments On Reviewer Discussion:**

The author response addressed the reviewers' questions. One of the main concerns was that the paper is not sufficiently novel compared to the prior work of Joseph and Yu. The author response addressed this concern and noted that the techniques are quite different.

---

### Decision · Program_Chairs · 2025-01-22

Accept (Poster)